# The Spike of Concern—The Novel Variants of SARS-CoV-2

**DOI:** 10.3390/v13061002

**Published:** 2021-05-27

**Authors:** Anna Winger, Thomas Caspari

**Affiliations:** 1Faculty of Pharmacy, Paracelsus Medical University, A-5020 Salzburg, Austria; anna.winger@stud.pmu.ac.at; 2Faculty of Medicine, Paracelsus Medical University, A-5020 Salzburg, Austria

**Keywords:** B.1.1.298, B.1.1.7, Marseille-4, B.1.429, B.1.427, B1.526, P.1, B.1.1.28, P.2, B.1.351, B.1.617

## Abstract

The high sequence identity of the first SARS-CoV-2 samples collected in December 2019 at Wuhan did not foretell the emergence of novel variants in the United Kingdom, North and South America, India, or South Africa that drive the current waves of the pandemic. The viral spike receptor possesses two surface areas of high mutagenic plasticity: the supersite in its N-terminal domain (NTD) that is recognised by all anti-NTD antibodies and its receptor binding domain (RBD) where 17 residues make contact with the human Ace2 protein (angiotensin I converting enzyme 2) and many neutralising antibodies bind. While NTD mutations appear at first glance very diverse, they converge on the structure of the supersite. The mutations within the RBD, on the other hand, hone in on only a small number of key sites (K417, L452, E484, N501) that are allosteric control points enabling spike to escape neutralising antibodies while maintaining or even gaining Ace2-binding activity. The D614G mutation is the hallmark of all variants, as it promotes viral spread by increasing the number of open spike protomers in the homo-trimeric receptor complex. This review discusses the recent spike mutations as well as their evolution.

## 1. Introduction

When the first SARS-CoV-2 patients were tested in December 2019 in Wuhan, China, the sequences of the single-stranded RNA virus were almost identical (≥99.9%), defining the new sequence family known as the L clade (reference sequence: EPI_ISL_402123) [1,2,3]. By June 2020, barely six months into the pandemic, the L clade dwindled to only 7% outside of Asia, and the dominating sequence group became the rapidly evolving G clade harbouring, amongst other changes, the D614G mutation in the viral spike protein (base change: A23403G, reference sequence: EPI_ISL_450201) [4]. Each SARS-CoV-2 virus carries approximately 90 homo-trimeric spike receptors in its membrane that vary in height between 9 nm and 12 nm [5]. They define spread and tropism of the virus as well as its ability to evade the immune system, as most neutralising antibodies bind to the viral receptor. Spike interacts with the human membrane protein angiotensin I converting enzyme 2 (hAce2) [6] that is expressed in the small intestine, colon, duodenum, kidney, testis, gallbladder, heart, and in many other tissues and cells at a lower level [7]. Interestingly, hAce2 expression in the upper airways and the lung is limited to certain cell types, like the goblet cells in the nasal mucosa or type 2 alveolar cells in the lung [7]. All variants of concern driving the recent waves of the COVID-19 (coronavirus disease 2019) pandemic are descendants of the D614G strain. A variant of concern (VOC) is a mutated strain of the Wuhan virus (reference genome: NC_045512.2) [8] that possesses a higher transmissibility, causes a more severe disease progression, increases mortality, escapes antibody neutralization, and/or evades detection. Variants posing only a possible risk to public health are classified by the WHO as Variant of Interest (VOI), and less hazardous strains are classified as Variants under Monitoring. As of 11 May 2021, Variants of Concern are B.1.1.7 (UK), B.1.351 (South Africa), P.1 (Brazil), and B.1.617 (India); the sole Variant of Interest is B.1.429/B.1.427 (USA), and Variants under Monitoring are B.1.526 (USA) and P.2 (Brazil).

Here, we summarize and discuss the mutations in spike that emerged in the UK (B.1.1.7), in Brazil (P.1/B.1.1.28 & P.2), in the USA (B.1.429/B.1.427 & B1.526), in South Africa (B.1.351), in Denmark (B.1.1.298), and in India (B.1.617). We also explore the drivers behind the rapid accumulation of mutations in spike.

## 2. The Structure and Biology of Spike

The N-terminal S1-domain of spike contains the receptor binding domain (RBD, 319–541aa) where 17 amino acids make direct contact with hAce2 at the plasma membrane of the target cell (Figure 1a, Figure 4) [9]. Three loops (N1, N3, N5) within the N-terminal domain (NTD, 13–305aa) define the so-called supersite where all known anti-NTD antibodies bind. This supersite is a surface patch free of glycans that otherwise shield most of spike with the exception of the RBD [10,11] (Figure 1e). A unique furin cleavage site separates S1 from the C-terminal S2 membrane fusion domain. While spike proteins from other coronaviruses carry a single arginine (R) at this position, SARS-CoV-2 has a unique four-amino-acid insertion (681-PRRA-684) immediately upstream of R685. Experimental deletion of these four residues reduced viral replication in a human respiratory cell line and attenuated disease progression in animal models [12,13,14]. The insertion may therefore promote viral spread between humans, a conclusion in line with its requirement for transmission of the virus between ferrets [15]. In contrast to other coronaviruses, the extended furin cleavage site (680–685aa) is already cut while the viral particles are forming at the endoplasmatic reticulum of the infected cell [13]. Furin cleavage destabilises the firmly closed trimer, thereby triggering its transition to a semi-stable prefusion state in which the S1 subunits undergo swinging motions. They allow the RBD domains to oscillate between their closed and open states while S1 remains bound to S2 (Figure 1c,d) [13,16,17,18]. Cleavage of the S2’ site (808–820aa) is, by contrast, strictly coupled to hAce2 binding, as it releases (sheds) the S1 subunit from S2. The resulting S2 fragment then undergoes a significant conformational change in which the two heptad repeats (HR1, HR2) form the six-HB fusion domain, and the fusion peptide (FP) (788–806aa) inserts itself into the plasma membrane of the target cell (Figure 1a,f) [19].

## 3. The D614G Mutation in Spike Facilitates SARS-CoV-2 Evolution

The emergence of the new variants was preceded by the global rise of the D614G mutation in spike that was first detected in January 2020 independently in China and Germany, outperforming the initial Wuhan virus by April/May 2020 (Figure 2) [21]. This mutation defines the new G clades (G, GH, GR, GV) that displaced the original L clade. Clade GR (D614G + G204R in the nucleocapsid (N) protein; reference sequence: EPI_ISL_850687) is currently dominating Africa (41.1%), Asia (52.7%), Oceania (74.8%), and South America (66.8%), while GH (D614G + Q57H in the NS3 protein; reference sequence: EPI_ISL_861025) reigns in North America (59.0%). The two clades GR (35.5%) and GV (D614G + A222V in spike; reference sequence: EPI_ISL_724371) (34.6%) are co-dominant in Europe [22].

Although the substitution of the negatively charged aspartate-614 by the uncharged and smaller glycine (D614G) removes a salt bridge with lysine-854 (K854) and a bond with T859, both in another spike protomer of the trimer, the overall structural changes in spike itself are limited to a loop (620–640 aa) close to its furin site. This loop is stabilized in G614 spike, as it fits into a larger opening generated by the smaller glycine, but is disordered in the D614 wild-type protein (Figure 3) [23]. Despite the small change, the fitness advantages for the virus are profound. While the wild-type trimer opens on average only one RBD, the G614 trimer opens two or even all three RBDs. Although G614 resides at a fair distance from the RBD, it affects the hAce2 binding site through an allosteric link with T500 [21,23,24]. This increases transmission of the virus especially at low viral loads [25]. Any gain in hAce2 binding is of high significance for viral spread, as only a small number of viruses (≤10) are passed on between people or test animals (i.e., SARS-CoV-2 has a small bottleneck of infection) [26,27]. Furthermore, the D614G mutation reduces furin cleavage, thereby lowering the risk of premature S1 shedding, and it enhances thermal stability of spike. Combined, these changes increase the viral load in the upper airways (nose and trachea) but not in the lungs [28,29]. This fitness gain explains why the G614 virus strongly outperforms the D614 virus in competition experiments using hamsters or ferrets. After inoculation of the test animals with an equal ratio of either strain, over 90% of the transmitted viruses carried the G614 spike [25]. While inter-person transmission becomes therefore more likely, neither disease progression nor neutralization by anti-spike antibodies are significantly affected by the D614G mutation [30,31].

### 3.1. The UK Variant B.1.1.7 (VOC202012/01, 20I/501Y.V1)

The variant B.1.1.7 (clade GR) emerged in late September 2020 in England (Figure 2), harbouring three deletions and seven replacements, including D614G, in spike (Table 1) [32]. Although B.1.1.7 is effectively neutralised after two doses of the RNA BioNTech/Pfizer vaccine BNT162b2 [31,33], a recent case-controlled study from Israel indicates two time windows, two weeks after the first dose and one week after the second dose of BNT162b2, where vaccinees can be reinfected by B.1.1.7 [34]. Transmission of this variant is 43% to 90% higher compared to the Wuhan strain [35]. This is probably the result of an increased number of open RBDs due to the D614G mutation combined with the N501Y replacement in the hAce2 binding site (Figure 4 and Figure 5) (Table 2). N501 resides within a field of hAce2 contacts at the rim of the concave binding site (Figure 5c), where it forms one hydrogen bond to tyrosine 41 in hAce2 (Figure 4). A ten-fold affinity gain is brought about by the N501Y mutation as Y501 interacts through a strong, aromatic stacking interaction with Y41 and forms two new hydrogen bonds with D38 and K353 in hAce2 [36]. Moreover, Y501 destabilises the RBD-down conformation, thereby adding to the D614G effect of more open RBDs [37]. An in-silico study revealed a key allosteric role of N501 together with E406, N439, and K417 as effector centres for long-range interactions running through S1, promoting its swinging motion after furin cleavage [18]. The increased affinity pushes the B.1.1.7 virus to higher concentrations in the upper airways in a hamster model and in human primary airway epithelial cells [38]. Intriguingly, the N501Y mutation evolved independently in the Brazilian P.1 and South African B.1.351 variants (Table 1) and was spontaneously selected when the Wuhan virus was repeatedly propagated in BALB/c mice [39]. The fitness gain indicated by the convergent evolution of N501Y is supported in the mouse model, where Y501 promotes a higher infectivity in the lung, enhances morbidity in the case of obesity, and pronounces interstitial pneumonia [39,40]. Although a higher hazard of human death (42–82%) was initially found in a community study of B.1.1.7 cases in the UK [41], this was later not confirmed by a cohort study in London [42]. While neutralisation of B.1.1.7 was not significantly reduced in vaccinated people, eight out of ten tested anti-NTD monoclonal antibodies failed to bind to B.1.1.7 spike in vitro [43]. The loss of neutralisation by the latter antibodies was attributed to the deletion of Y144 in loop N3 of the supersite (Figure 1). This conclusion is supported by a 3D-model of B.1.1.7 spike in which the topology of loop N3 (140–158aa) changes (Figure 6b). The other two deletions in the NTD (ΔH69, ΔV70) were reported to alter loop N2 (69 to 76aa), pulling it closer to the NTD [44]. Although the P681H mutation at the furin cleavage site initially raised much interest, it has no significant impact on viral fitness and was only found to slightly increase S1/S2 cleavage in cell culture [45]. What is, however, worth noting is the convergent mutation of P681 to an arginine (P681R) in the Indian B.1.617 Spike (Table 1), implying a functional role for a positive charge at this position (histidine can also be positively charged at a lower pH value).

### 3.2. The Brazilian P.1 (20J/501Y.V3) Variant

The P.1 variant (a descendent of B.1.1.28) was first detected in Manaus, Amazonas State, on 6 December 2020 (Figure 2). Evolutionary modelling suggests that the P.1 variant emerged one month prior to its detection at the beginning of November 2020, after a period of intense molecular selection (where this selection took place is unknown). P.1 Spike contains 12 amino acid changes of which K417T, E484K, and N501Y are key positions with a high degree of structural and energetic plasticity that can accommodate novel residues with altered Ace2-interaction potentials (Table 1) [18,52].

Although lysine-417 (K417) lies outside of the core hAce2 binding region (437–508aa) at the side of the concave RBD surface, it forms a salt bridge with D30 in hAce2 [9] (Figure 4). Both the K417T and K417N mutations abolish this bond and are therefore predicted to have a lower hAce2 affinity while helping to evade neutralizing antibodies [56,57]. The main impact of the K417N mutation seems to be its ability to destabilize the RBD-down conformation, thereby increasing the propensity of the open configuration [37]. Whether this applies also to the K417T mutation is not yet clear, but it is likely, as K417 is a major site in the RBD with long-range, inter-molecular links throughout spike [18] (Table 2).

Glutamate-484 (E484) forms a polar link with K31 in hAce2 despite its position at the outer rim of the RBD [9] (Figure 4 and Figure 5). E484 also stabilizes the RBD-down conformation by contacting the N343-glycan and F490 in the neighbouring RBD [37]. Since both interactions are absent in the E484K mutant, the S1 movements favour the RBD-up conformation.

The E484K mutation attracted much attention due to its ability to prevent binding of neutralizing antibodies [58,59,60,61] and to be selected as an escape mutation in the presence of neutralizing antibodies or plasma from immune humans in vitro [59,62]. Intriguingly, recent data from England and Wales (February 2021) show the accumulation of E484K in the B.1.1.7 background, suggesting a selection of this mutation in the response to the vaccination program [43]. E484K is also the only shared mutation of both Brazilian variants (Table 1), which possess a similar resistance to neutralization after two doses of the BioNTech/Pfizer (BNT162b2) or two doses of the Moderna (mRNA-1273) vaccine [31]. Taken together, mutation of E484 is important to monitor, as it combines the advantages of conformational changes with the escape from antibodies. The importance of this position is further underscored by the convergent appearance of the E484Q mutation in the Indian B.1.617 variant (Table 1).

### 3.3. The North American Variants B.1.429 and B.1.526

The variant B.1.429 (CAL.20C/B.1.427) was originally detected in California, United States, in May 2020 [49] (Figure 2). Its spike protein carries the L452R replacement at the outer rim of its hAce2 binding site (Figure 4b). While L452 does not make direct contact with hAce2, its neighbour Y453 is involved in receptor binding [9]. L452 forms with L492 and F490 a hydrophobic patch at the bottom of the hAce2 binding site that is disrupted by the positively charged lysine, thereby preventing the binding of neutralising antibodies (Figure 5f) [48,55]. This may explain why L452R emerged independently in the Indian B.1.617 variant (Table 1). The L452R mutation increases hAce2 affinity only slightly but results in much higher viral replication in non-human veroE6 cells [65] suggesting a role for this site in animal reservoirs like the mink [66]. This is supported by the mutation of Y453 in the Danish B.1.1.298 strain that transferred reversibly between humans and mink.

In addition to this important change, B.1.429 Spike has a dramatic rearrangement at its supersite where loop N5 (245–264aa) is pushed away from the surface (Figure 6c). This structural change is probably caused by a novel disulfide bond between C136 and the W152C mutation in loop N3 (140–158aa). This new disulfide bond forms since the original partner of C136, cysteine 15, is no longer available due to an aberrant cleavage of the signal peptide immediately after C15. This aberrant modification is caused by the S13I mutation enabling B.1.429 to escape many anti-NTD antibodies [48].

In late December 2020, the strain B.1.526 with the E484K mutation in Spike was identified in New York [50]. Although the B.1.526 NTD harbours with L5F, T95I, and D253G, three mutations which are distinct from B.1.429, the 3D model of B.1.526 spike predicts a similar structural rearrangement of loop N5 (245–264aa) (Figure 6d). This aberration may be linked with the D253G replacement in this loop. Hence, both strains may have independently evolved a similar structural change in loop N5 to escape anti-NTD antibodies.

### 3.4. The South African B.1.351 Variant

In October 2020, the strain B.1.351 (B.1.351-V1) emerged in South Africa. Its Spike protein shares the three RBD mutations at the key residues K417, E484, and N501 (K417N, E484K, N501Y) with the Brazilian P.1 strain (please note the K417 is replaced by an asparagine in B.1.351 but a threonine in P.1) (Figure 2, Table 1). B.1.351 Spike contains also a consecutive deletion of the three amino acids L242, A243, and L244 immediately N-terminal to loop N5 (245–264aa) as well as the replacements D80A, D215G, and A701V [54]. In November 2020, two further variants of B.1.351 were detected. Variant B.1.351-V2 carries one additional N-terminal mutation, L18F, which is also present in the Brazilian P.1 virus, and variant B.1.351-V3, which harbours the R246I mutation in loop N5 but lacks the D215G replacement (Table 1) [31,54].

The 3D-models of B1.351-V1 and B.1.351-V2 spike show a similar outwards movement of loop N5 as observed in the models of the two North American spike proteins B.1.429 and B.1.526 (Figure 6e,f). This structural rearrangement may be caused by the deletion of the three amino acids L242, A243, and L244 just N-terminal of loop N5. The displacement of loop N5 prevents binding of the anti-NTD monoclonal antibody 159 [67] and of other anti-NTD antibodies [61]. Interestingly, the replacement of arginine-246 by an isoleucine (R246I) at the beginning of loop N5 in Spike B.1.351-V3 suppresses this structural aberration (Figure 6g). Whether this restores neutralization by anti-NTD antibodies is currently unknown.

While the affinity of B.1.351-V1 for hAce2 increases 19-fold due to the three RBD mutations and D614G, its neutralization by serum from convalescent patients is on average 13-fold lower and is also reduced in serum from people vaccinated with Oxford-AstraZeneca AZD1222 (9-fold) or immunized with Pfizer-BioNTech BNT162b2 (7.6-fold) [67]. A second study reported a more than 70-fold reduction in neutralization after immunization with BNT162b2 or mRNA-1273 for all three variants, an effect attributed to the three RBD mutations, with the strongest reduction (≥90-fold) measured for B.1.351-V2 [31]. The mounting evidence of impaired neutralization of the B.1.351 variants after recovery from an infection or after vaccination raises the concern of possible reinfections by the Brazilian SARS-CoV-2 viruses [68].

### 3.5. The Danish B.1.1.298 Variant

The first sequences of the Danish B.1.1.298 variant were obtained in June 2020 from a small number of infected workers at a mink farm. The so-called cluster 5 strain contains the same two N-terminal deletions (ΔH69, ΔV70) as the UK B.1.1.7 Spike as well as the three mutations Y453F, I692V, and M1229I in addition to D614G (Table 1) [46]. Y453 forms an electrostatic interaction with H34 in hAce2 and lies at the bottom of the concave binding site next to L452 (mutated in USA B.1.429 and Indian B.1.617 Spike) (Figure 4a and Figure 5f). The Y453F mutation arose several times independently in different sub-lineages, was found as an escape mutant in in vitro experiments [69], and increases hAce2 affinity four-fold [70] (Table 2). The mutation may be linked with viral replication in minks and the repeated transition from humans to mink and back [66,71]. Unlike the other SARS-CoV-2 variants, transmission of B.1.1.298 was limited to a few hundred people mainly in the vicinity of mink facilities. Although it is no variant of concern, the strain is of significance, as is demonstrates the ability of SARS-CoV-2 to reversibly cross species boundaries and to possess a high mutation rate in an animal reservoir (in this case, mink) [66].

### 3.6. The Indian B.1.617 Variant

B.1.617 is currently not a defined variant, as it consolidates a group of sequence clusters within clade G that share the common signature mutations: G142D, L452R, E484Q, D614G, and P681R (Table 1) as well the silent mutation D111D [55]. The first sequence cluster was noticed in October 2020 in India (Figure 2). It carries the Spike mutations T19R, G142D, L454R, E484Q, D614G, P681R, and D950N. In January 2021, the mutation Q1071H emerged, giving rise to three new sub-clusters (B.1.617:A, B.1.617:B, B.1.617C) [55]. There are seven additional mutations (T19R, K77T, T95I, E154K, N440K, T478K, H1101D) that occur with varying frequencies in the B.1.617 sequence background [55]. The European Centre for Disease Prevention and Control (ECDC) lists currently the following sub-variants: B.1.617.1 was detected in India in December 2020 (L452R, E484Q, D614G, P681R, Q1071H (some viruses also carry V382L)), B.1.617.2 was initially identified in India in December 2020 but is now on the rise in the UK (T19R, Δ157–158, L452R, T478K, D614G, P681R, D950N), and the rare variant B.1.617.3 (T19R, Δ157–158, L452R, E484Q, D614G, P681R, D950N) that was found in February 2021 in India [72]. While the data related to B.1.617 are still scarce, experiments with B.1.617.1 in the hamster model indicate a higher pathogenicity compared to the D614G variant [73]. Antibodies and sera from vaccinated people are, however, able to neutralise a pseudotyped VSV virus carrying the B.1.617 spike with the mutations R21T, E154K, Q218H, L452R, E484Q, D614G, P681R, and H1101D (GISAID Accession ID: PI_ISL_1360382) [74].

Taken together, these findings suggest a highly dynamic evolution of the B.1.617 spike protein with a strong regional pattern.

### 3.7. The Evolution of the New Strains

An interesting hallmark of the new SARS-CoV-2 variants is the co-evolution of mutation clusters in Spike that advance viral fitness. The evolution rate of spike is with 5 × 10^−3^ substitutions/site/year (i.e., average number of substitutions that become fixed per year), three times higher than the evolution rate across the entire SARS-CoV-2 genome (1.6 × 10^−3^ substitutions/site/year) but still within the range of other betacoronaviruses [5,8]. The mutation rate (i.e., number of substitutions per site per replication cycle) is within 10^−6^ mutations/site/replication cycle, high enough to mutate on average every amino acid in spike at least once in one patient [5]. The GISAID database illustrates this, as it currently lists one mutation for almost every Spike amino acid. While most mutations occur only in a few sequences, only a small number of changes are positively selected, as they convey benefits with regard to immune evasion, viral fitness, and/or hAce2 binding, thereby contributing to the evolution rate. For this to happen, the virus needs to replicate for a longer period in one host under selection pressure. It is thought that novel variants evolve in immune-compromised patients suffering from long COVID and/or in an animal reservoir, like the mink, where the virus undergoes extensive evolution before returning to humans. For example, a high in-host evolution rate was found over a period of 70 days in a cancer patient [75], and the spontaneous appearance of the RBD mutations E484K and N501Y were observed 75 days and 128 days post-diagnosis in a patient suffering from the immune disorder antiphospholipid syndrome [76]. Interestingly, in both cases, short deletions in the supersite loop N3 were found. Given the high expression level of Ace2 in the gastrointestinal system [7], it is worth noting that active replicating virus was found in anal samples [77], suggesting that the gastrointestinal system may provide an evolution refuge for SARS-CoV-2.

An example of a spill over from an animal reservoir is the Marseille-4 variant (Table 1) that appeared suddenly in France at the end of the first SARS-CoV-2 wave in July 2020 (Figure 2) in an area close to a mink farm [47].

Figure 7 illustrates the leading theme of the evolution of the hAce2 binding site of spike. Allosteric key positions, like E484 or N501, possess sufficient structural freedom to accommodate new residues without significant affinity loss (sometimes even affinity gains, i.e., N501Y) while changing the surface of the binding site such that neutralising antibodies fail to bind. This was shown for the E484K mutation that effects the epitope of the 80R antibody and the N501Y mutation that impairs binding of 80R and a second neutralising antibody, M396 (Figure 7) [9].

In conclusion, the adaptability of the viral genome and the continuous appearance of escape mutations, as illustrated by the recent appearance of N439K in Scotland [78] or the dynamic evolution of B.1.617, suggest that SARS-CoV-2 has the potential to become a seasonal virus like influenza with the need of re-adjusting the vaccines.

## Figures and Tables

**Figure 1 viruses-13-01002-f001:**
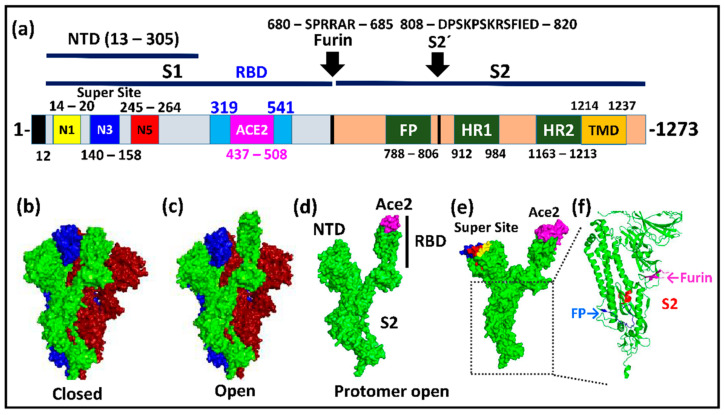
Domain Organization of the SARS-CoV-2 spike protein and structural features. (**a**) Domain organization of spike. Signal peptide: 1–12aa; S1-domain: 13–685aa; S2-domain: 686–1273aa; N-terminal domain (NTD): 13–305aa; Supersite loops: N1:14–20aa; N3: 140–158aa; N5: 245–264aa; Receptor binding domain (RBD): 319–541aa; Receptor binding motif (ACE2): 437–508aa; Furin cleavage sequence: 680–685aa; Fusion peptide (FP): 788–806aa; Heptad repeat region 1 (HR1): 912–984aa; Heptad repeat region 2 (HR2): 1163–1213aa; Transmembrane domain (TMD): 1214–1237aa; Cytoplasmic domain: 1238–1273aa; (**b**) Surface of the closed spike trimer (PDB: 6VXX); (**c**) Surface of the open spike trimer—one protomer open (green) (PDB: 6VYB); (**d**) Open protomer from (**c**) with the Ace2 binding motif (magenta) and the RBD both indicated; (**e**) Surface of the open protomer with the three loops of the N-terminal supersite (N1: yellow; N3: blue; N5: red) and the Ace2 binding site (magenta) highlighted (PDB: 7DF4); (**f**) Cartoon rendering of the lower S2 section. The Furin loop is not fully visible due to its high mobility (magenta). The S2 cleavage site is shown in red and the fusion peptide (FP) in blue (PDB: 7DF4). Domain annotation according to [19]. Loops N1, N3, and N5 annotation as in [10]. Visualisation with Polyview-3D [20].

**Figure 2 viruses-13-01002-f002:**
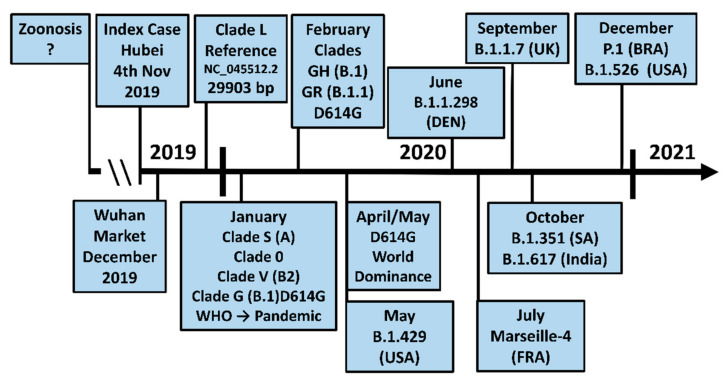
Timeline of the SARS-CoV-2 pandemic and the emergence of the variants. See text for further details.

**Figure 3 viruses-13-01002-f003:**
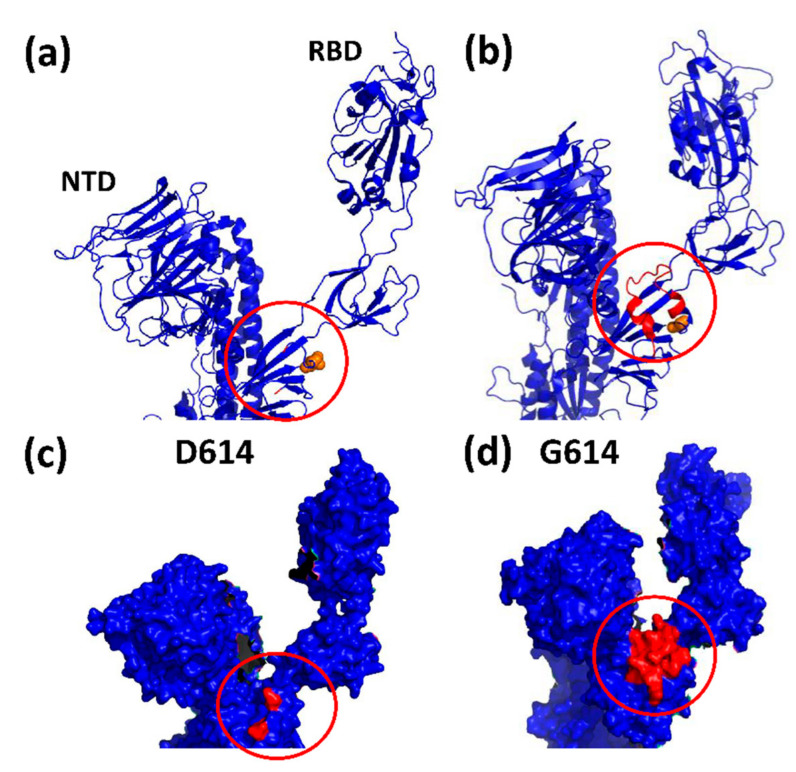
Structural impact of the D614G mutation. (**a**) Cartoon rendering of the open spike protomer (only top half is shown) with an aspartate at position 614 (D614, orange spheres); the disordered 620–640 loop (red circle) is absent from the structure; NTD (N-terminal S1 domain); RBD (receptor binding domain in S1) (PDB: 7KRR); (**b**) Cartoon rendering of the open spike protomer with a glycine at position 614 (G614, orange spheres) (PDB: 7DK3); the ordered 620–640 loop (red) is now visible; (**c**) Surface rendering of spike D614; (**d**) Surface rendering of spike G614. Visualisation with PyMOL (Molecular Graphics System, Version 2.0 Schrödinger, LLC).

**Figure 4 viruses-13-01002-f004:**
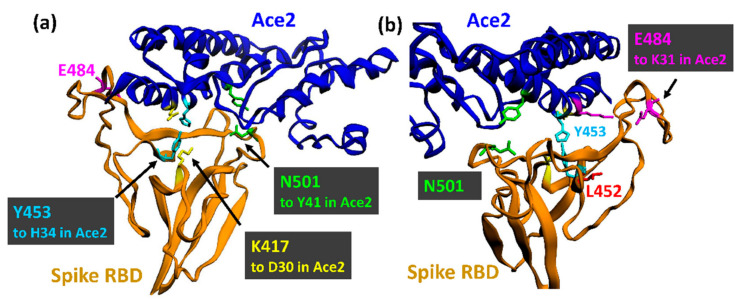
The receptor binding motif of spike (RBD). (**a**) View into the binding site. The key amino acids which are mutated in the variants (Table 1) are highlighted. The binding partners in hAce2 are shown in the same colour. Please note that L452 (red) is next to Y453 (light blue) but facing away from the binding site. E484 (magenta) is positioned at the rim of the binding site. (**b**) The backside of the binding site. The polar link between E484 with K31 in hAce2 and L452 are now visible. The other 14 residues making contact with hAce2 (G446, Y449, L455, F456, A475, F486, N487, Y489, Q493, G496, Q498, T500, G502, Y505) are not shown. hAce2 (blue): amino acids 19–99 and 312–395 are shown. Spike (orange): amino acids 393–516 are shown. The image was rendered with Ezmol 2.1 [63] using the structure PDB 7DF4.

**Figure 5 viruses-13-01002-f005:**
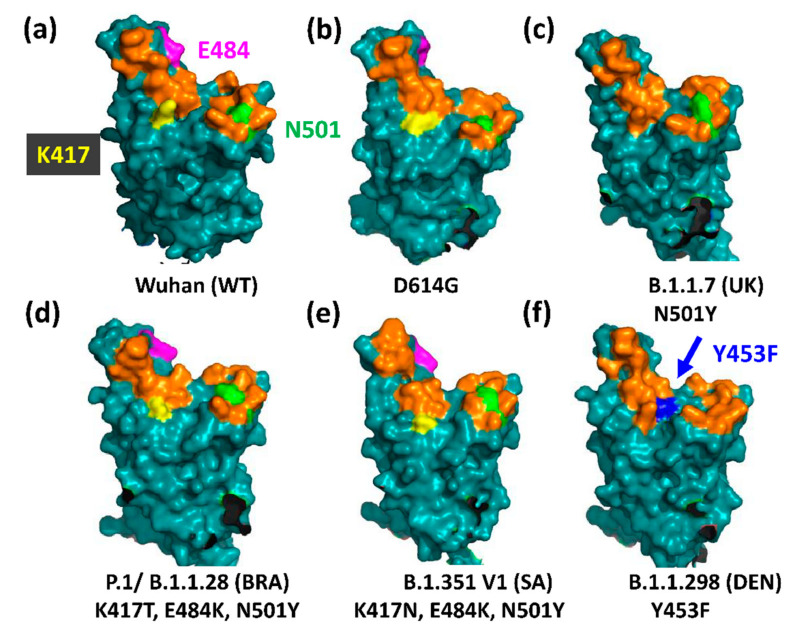
Surface structure of the hAce2 binding site of Spike and its variants. (**a**) The Wuhan strain. The 17 amino acids making direct contact with hAce2 are highlighted in orange, K417 in yellow, N501 in green and E484 in magenta [9]; (**b**) D614G (PDB: 7BNN); (**c**) B.1.1.7 (UK) (PDB: 7LWT); (**d**) P.1/B.1.1.28 (Brazil) (PDB: 7LWW); (**e**) B.1351 V1 (South Africa) (PDB: 7LYQ); (**f**) B.1.1.298 (Denmark) (PDB: 7LWO); only the Wuhan structure was modelled with Phyre2 (Protein Homology/analogy Recognition Engine V 2.0) and visualised with EzMol2.1 [63,64]; the other structures are visualisations of the published structural data sets with the complete coverage of the RBD (i.e., no unstructured regions). Visualisation with PyMOL (Molecular Graphics System, Version 2.0 Schrödinger, LLC). Wuhan spike protein sequence: UniProt P0DTC2.

**Figure 6 viruses-13-01002-f006:**
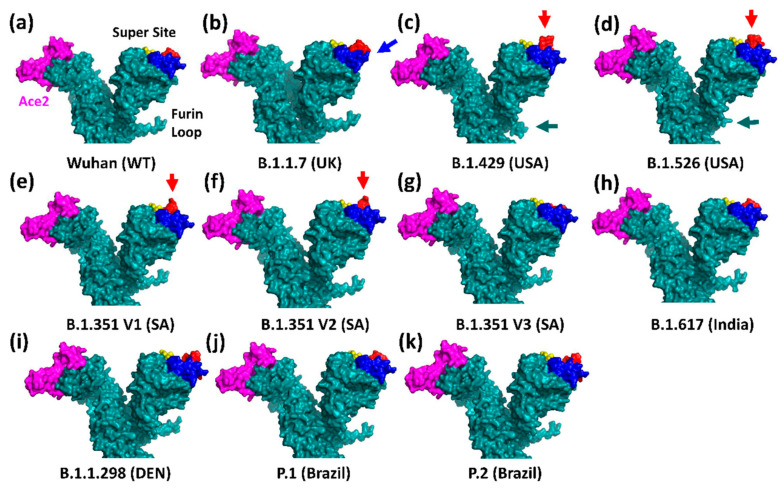
Models of the protein surfaces of wild-type Spike and its variants with the three loops of the super site, the hAce2 binding site, and the furin loop indicated. (**a**) The Wuhan wild-type spike (loops N1 (14–20aa): yellow; N3 (140–158aa): blue; N5 (245–264aa): red; hAce2 binding site (magenta); furin loop (green arrow)); (**b**) B.1.1.7 (UK); (**c**) B.1.429 (USA); (**d**) B.1.526 (USA); (**e**–**g**) sub-variants V1, V2, and V3 of B.1.351 (South Africa); (**h**) B.1.617 (India) (please note the model contains only the mutations listed in Table 1); (**i**) B.1.1.298 (Denmark); (**j**) P.1 (Brazil); (**k**) P.2 (Brazil). Since three-D structure are not yet published for all mutated Spike proteins, models were generated with the mutated amino acid sequences of Spike using the Phyre2 (Protein Homology/analogy Recognition Engine V 2.0) server. The resulting structure files were then rendered with PyMOL (Molecular Graphics System, Version 2.0 Schrödinger, LLC). Arrows indicate the predicted structural changes in loop N3 (blue) or loop N5 (red). Please note that the furin loop is modelled differently in both USA variants compared to all other variants. The reason for this is as yet unknown.

**Figure 7 viruses-13-01002-f007:**
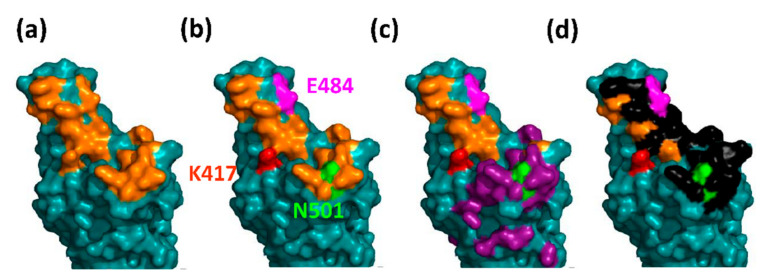
The structural plasticity of the hAce2 binding site in Spike. (**a**) The surface of the hAce2 binding site of the Wuhan strain is shown in all subpanels. The 17 amino acids making direct contact with hAce2 are highlighted in orange; (**b**) the positions of the mutations K417T/N (red), E484K (magenta), and N501Y (green) present in the Brazilian P.1 and the South African B.1.351 Spike proteins are highlighted; (**c**) the epitope of the neutralising antibody M369 is shown in dark purple; (**d**) the epitope of the neutralising antibody 80R is shown in black [9]. Please note that N501 resides within both epitopes. Hence, its mutation enhances hAce2 affinity while simultaneously impairing antibody binding. Images were rendered with PyMOL (Molecular Graphics System, Version 2.0 Schrödinger, LLC); the structure file was generated with Phyre2 (Protein Homology/analogy Recognition Engine V 2.0) using the Wuhan Spike protein sequence: UniProt P0DTC2.

**Table 1 viruses-13-01002-t001:** SARS-CoV-2 variants.

SARS-CoV-2 Variant	Other S1 Mutations	RBD Mutations	Furin	S2 Mutations
B.1.1.7 (UK) [32]	ΔH69, ΔV70, ΔY144, A570D, D614G	**N501Y**	P681H	T716I, S982A, D1118H
B.1.1.298 (DEN) [46]	ΔH69, ΔV70, D614G	**Y453F**		I692V, M1229I
Marseille-4 (20A.EU2) [47]	S447N			
B.1.429/B.1.427 (USA) [48,49]	S13I, W152C, D614G	L452R		
B.1.526 (USA) [50]	L5F, T95I, D253G, D614G	E484K		A701V
P.1/B.1.1.28 (BRA) [31,51,52]	L18F, T20N, P26S, D138Y, R190S, D614G, H655Y	**K417T**, E484K, **N501Y**		T1027I, V1176F *
P.2 (BRA) [53]	D614G	E484K		V1176F
B.1.351 v1 (SA) [54]	D80A, D215G, ΔL242, ΔA243, ΔL244, D614G	**K417N**, E484K, **N501Y**		A701V
B.1.351 v2 (SA) [31]	L18F, D80A, D215G, ΔL242, ΔA243, ΔL244, D614G	**K417N**, E484K, **N501Y**		A701V
B.1.351 v1 (SA) [31]	D80A, ΔL242, ΔA243, ΔL244, R246I, D614G	**K417N**, E484K, **N501Y**		A701V
B.1.617 (India) [55]	G142D, D614G	E484Q, L452R	P681R	

Bold and underlined residues make direct contact with Ace2 [9]. * The additional mutation, V1176F, for P.1 is listed in [31].

**Table 2 viruses-13-01002-t002:** Consequences of the amino acid replacements.

Replacement	Conformation	hAce2 Affinity	Neutralization	Other
S13I				Aberrant cleavage of the signal peptide
ΔH69, ΔV70	N2 loop in supersite		Anti-NTD antibodies ↓	Found in mink
ΔY144	N3 loop in supersite		Anti-NTD antibodies ↓	
W152C	Displacement loop N5		Anti-NTD antibodies ↓	New disulfide bond C136-C152
ΔL242, ΔA243, ΔL244	Displacement loop N5		Anti-NTD antibodies ↓	
K417T/N	More RBDs open	Reduced		S1 allosteric centre
L452K	Hydrophobic patchinterrupted		Anti-RBD antibodies ↓	S1 allosteric centre
Y453F		4-fold increased		Escape mutation, found in mink
E484K	More RBDs open	Reduced	Anti-RBD antibodies ↓	Escape mutation
N501Y	More RBDs open	10-fold increased	Anti-RBD antibodies ↓	Selected when virus is adapted in mouse, S1 allosteric centre
D614G	More RBDs open	Reduced	Not affected	Increased stability
P681H/R				Impact on furin cleavage

↓: reduced antibody binding.

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
