# Peer review of "The Spike of Concern—The Novel Variants of SARS-CoV-2"

_viruses, 2021, doi:10.3390/v13061002_

Round 1

Reviewer 1 Report

The review by Winger & Caspari summarizes some of the major SARS-CoV-2 Spike variants appeared since December 2019. It is neatly written and condenses some relevant aspects of the virus and Spike biology with an effort to simplify the link between the appearance of the variants, their impact on the spike structure and their consequences on vaccines’ development. The final section concludes on the evolutionary aspects of the emergence of the new strains.

The major drawback of this review is the rendering of the figures and tables which are of the outmost importance.

Figure 1:  please increase a bit the space from panel (a) and the below panels (b-f) as it is the figure appears to be crowded. The color scheme of (b-f) is a bit confusing. For example, panel (d) what is in green? Panel (e) there are two regions in green – what are they?  Please revise.

Figure 3: the magenta and red on the cartoon representation of the molecule are not clearly distinguishable – would it be possible to use, for example an orange color for G614?

Table 1. Although informative it looks to crowded – would it be possible to slightly decrease the font size and use vertical lines delimiting each column?

Figure 4: please use the white background and mention the software used for depicting/rendering. Also, the Authors should specify that this is a cartoon representation with some residues in stick.

Table 2: see comment as for Table 1

Figure 5:  For clarity, the panels should have a white background – also it is not clear if the authors are displaying/rendering the protein surface or else. The coloring is clear but what they are displaying not. I would suggest for this type of figures to use Pymol.

In the corresponding legend the Authors comment that the structure was modelled with Phyre2 …..why the need of modelling? The whole domain or the single mutation on a previously determined cryo-EM or X-ray structure? This part is not clear and must be clarified the use of Phyre2 modelling in the context of the experimentally determined structures (either by cryo-EM or X-ray).

Figure 6: please introduce some space between the panels (a-d) and those below (e-h). As for Figure 5 it is not clear if this is a surface and what type of surface. Again, I would suggest to use Pymol for this type of figures as widely use in the field and considered one of the best software to display molecular structures.

Minor revisions:

Page 3 – line 93:  ‘in the N protein’: the authors have not defined yet what is the N protein …is it the SARS-CoV-2 nucleoprotein? It should be clarified.

Page 7 – line 252:  please correct ‘N501Y:’ with ‘N501Y;’

Page 8 – line 282: the ordering of presentation of panels in Figure 6 is not sequential within the text. This generates some confusion. For example, Figure 6e is mentioned at page 4 line 145 whereas Figure 6b at page 5 line 164…also where is mentioned in the text Figure 6a? The ordering of the panels should follow the mentioning within the text as much as possible.

Page 10 – line 355: please correct ‘occure’ with ‘occur’

Page 11 – line 382: please correct ‘influenca’ with ‘influenza’

bioRxiv references should be updated with final publications when possible.

Author Response

Reviewer 1

Comment 1

Figure 1:  please increase a bit the space from panel (a) and the below panels (b-f) as it is the figure appears to be crowded. The colour scheme of (b-f) is a bit confusing. For example, panel (d) what is in green? Panel (e) there are two regions in green – what are they?  Please revise.

Response

The space between panel (a) and the below panels (b-f) has been increased. The colour scheme in panel (d) has been simplified to show that the green protomer is the same green protomer as in panel (c) but with the Ace2 binding site (magenta) and the RBD both indicated. Panel (e) has also been simplified. The three loops of the super site are shown in yellow (N1, 14-20aa), blue (N3, 140-158) and red (N5, 245-264), and the Ace2 binding site (437-508aa) in magenta. Panel (f) shows now only the lower section of the Spike protein with S´, the Furin site and the fusion peptide. These structures are now more prominent in the figure. The figure legend was updated accordingly.

Comment 2

Figure 3: the magenta and red on the cartoon representation of the molecule are not clearly distinguishable – would it be possible to use, for example an orange color for G614?

Response

Figure 3 has bow replaced using Pymol rendered images to improve clarity.

Comment 3

Table 1. Although informative it looks to crowded – would it be possible to slightly decrease the font size and use vertical lines delimiting each column?

Response

The font size of Table 1 has been reduced to 9 points and vertical lines were included.

Comment 4

Figure 4: please use the white background and mention the software used for depicting/rendering. Also, the Authors should specify that this is a cartoon representation with some residues in stick.

Response

The background colour in Figure 4 has been changed to white. We have removed the stick representation of the other contact amino acids in Spike for clarity (they are now only listed in the figure legend). The software has been included in the legend with reference.

Comment 5

Table 2: see comment as for Table 1

Response

The font size of Table 2 has been reduced to 9 points and vertical lines were included.

Comment 6

Figure 5:  For clarity, the panels should have a white background – also it is not clear if the authors are displaying/rendering the protein surface or else. The coloring is clear but what they are displaying not. I would suggest for this type of figures to use Pymol.

Response

We used Pymol to revise Figure 3, 5, 6 and 7. It is now stated that the surface area was rendered.

Comment 7

In the corresponding legend the Authors comment that the structure was modelled with Phyre2 …..why the need of modelling? The whole domain or the single mutation on a previously determined cryo-EM or X-ray structure? This part is not clear and must be clarified the use of Phyre2 modelling in the context of the experimentally determined structures (either by cryo-EM or X-ray).

Response

The Phyre2 server was used to model the images shown in Figure 6 since cryo-EM or X-ray structures are not yet available for all mutated Spike proteins. Modelling on the Phyre2 server is based on homology to the published Spike structures while taking account of possible changes caused by the amino acid replacements in the variants. The available published variant structures have now been included in the revised Figure 5. In some cryo-EM structures, important loops in the Ace2 binding site and/or in the supersite (e.g. N1, N3, H5 in PDB-ID: 7LWT of UK B.1.1.7) are unfortunately not fully resolved and therefore not displayable, hence the bypass via the Phyre2 server.

Comment 8

Figure 6: please introduce some space between the panels (a-d) and those below (e-h). As for Figure 5 it is not clear if this is a surface and what type of surface. Again, I would suggest to use Pymol for this type of figures as widely use in the field and considered one of the best software to display molecular structures.

Response

We have used Pymol for all images as suggested. Now all variants discussed in the text are shown. The figure legend states that the protein surfaces are depicted.

Minor revisions:

Comment 9

Page 3 – line 93:  ‘in the N protein’: the authors have not defined yet what is the N protein …is it the SARS-CoV-2 nucleoprotein? It should be clarified.

Response:

Yes it is the the nucleocapsid protein – the full name has now been included

Comment 10

Page 7 – line 252:  please correct ‘N501Y:’ with ‘N501Y;’

Response

Corrected

Comment 11

Page 8 – line 282: the ordering of presentation of panels in Figure 6 is not sequential within the text. This generates some confusion. For example, Figure 6e is mentioned at page 4 line 145 whereas Figure 6b at page 5 line 164…also where is mentioned in the text Figure 6a? The ordering of the panels should follow the mentioning within the text as much as possible.

Response

The ordering of the panels follows now the story line in the text.

Comment 12

Page 10 – line 355: please correct ‘occure’ with ‘occur’

Response

corrected

Comment 13

Page 11 – line 382: please correct ‘influenca’ with ‘influenza’

Response

Corrected

Comment 14

bioRxiv references should be updated with final publications when possible.

Response

The bioRxiv references that were meanwhile published are now updated

Reviewer 2 Report

The review is a comprehensive summary of the SARS-CoV2 S variants known till date. The illustrations are well done and enhance the readers' understanding. However, there are numerous typographical errors in the text. In addition, some sentence re-structuring is required to improve clarity. Some specific edits required are listed below:

  1. In Table 1, the second column heading should be 'Other S1 mutations' since RBD and Furin mutations are also S1 mutations.
  2. Line 135: For the sake of clarity, please refrain from referring to the 'UK' variant as the 'British' variant.
  3. Line 58: Please cite the following studies: Lau et al. Emerg. Microbes Infect. 2020; Peacock et al. Nature Biotechnology 2021.
  4. Please distinguish between 'variants of concern' and 'variants of interest' since not all variants discussed in this review are categorized as 'variants of concern' by WHO/CDC. 

Author Response

Reviewer 2

The typographical errors in the text were corrected and the long sentences re-structured to improve clarity.

Comment 1

In Table 1, the second column heading should be 'Other S1 mutations' since RBD and Furin mutations are also S1 mutations.

Response

Corrected

Comment 2

Line 135: For the sake of clarity, please refrain from referring to the 'UK' variant as the 'British' variant.

Response

Corrected

Comment 3

Line 58: Please cite the following studies: Lau et al. Emerg. Microbes Infect. 2020; Peacock et al. Nature Biotechnology 2021.

Response

Both references are now included. Included is: Peacock T et al., 2021 Nat Microbiology

Comment 4

Please distinguish between 'variants of concern' and 'variants of interest' since not all variants discussed in this review are categorized as 'variants of concern' by WHO/CDC. 

Response

Good point – done on page 1

Round 2

Reviewer 1 Report

The suggested revisions have been addressed by the authors.

Reviewer 2 Report

The authors have incorporated the changes recommended in the first round of review. The manuscript is more clear now and can be accepted after correcting minor typographical errors.